# Role of the Microbiota in Skin Neoplasms: New Therapeutic Horizons

**DOI:** 10.3390/microorganisms11102386

**Published:** 2023-09-25

**Authors:** Paola Savoia, Barbara Azzimonti, Roberta Rolla, Elisa Zavattaro

**Affiliations:** Department of Health Science, University of Eastern Piedmont, via Solaroli 17, 28100 Novara, Italy; barbara.azzimonti@med.uniupo.it (B.A.); roberta.rolla@med.uniupo.it (R.R.); elisa.zavattaro@med.uniupo.it (E.Z.)

**Keywords:** microbiota, gut–skin axis, skin cancer, melanoma, cutaneous carcinogenesis

## Abstract

The skin and the gut are regularly colonized by a variety of microorganisms capable of interacting with the immune system through their metabolites and influencing the balance between immune tolerance and inflammation. Alterations in the composition and diversity of the skin microbiota have been described in various cutaneous diseases, including skin cancer, and the actual function of the human microbiota in skin carcinogenesis, such as in progression and metastasis, is currently an active area of research. The role of Human Papilloma Virus (HPV) in the pathogenesis of squamous cell carcinoma is well consolidated, especially in chronically immunosuppressed patients. Furthermore, an imbalance between *Staphylococcus* spp., such as *Staphylococcus epidermidis* and *aureus*, has been found to be strongly related to the progression from actinic keratosis to squamous cell carcinoma and differently associated with various stages of the diseases in cutaneous T-cell lymphoma patients. Also, in melanoma patients, differences in microbiota have been related to dissimilar disease course and prognosis and may affect the effectiveness and tolerability of immune checkpoint inhibitors, which currently represent one of the best chances of a cure. From this point of view, acting on microbiota can be considered a possible therapeutic option for patients with advanced skin cancers, even if several issues are still open.

## 1. Introduction

The intestinal microbiota, i.e., the whole of microorganisms (bacteria, viruses, fungi, and protozoa) that colonize the gastro-enteric tract, has a broad and complex interaction with the immune system through metabolites able to influence the balance between the immune tolerance and the inflammatory state [1]. Also, it communicates with the skin, as an important regulator of the gastrointestinal–skin axis system (“gut–skin axis”) [1].

Recent studies have demonstrated, in fact, that the intestinal microbiota can influence the skin pathophysiology and its corresponding immune response through the cutaneous migration of microorganisms and their metabolites. These ones can enter the bloodstream, through a damaged gastrointestinal barrier and reach the skin, causing distant effects and contributing to the pathogenesis of numerous chronic inflammatory diseases, such as psoriasis and atopic dermatitis. Increasing evidence has also shown a putative role of microorganisms in the pathogenesis and progression of skin cancer, through both a direct influence on cell proliferation and death processes and indirect effects on host immunity and metabolism [2]. Moreover, microbiota may affect the response to immunotherapy and its tolerability in patients affected by skin malignancies [3].

Herein, we summarize the most recent publications on this attractive topic, addressing the potential therapeutic implications, but also possible limits, given the current knowledge.

## 2. The Human Microbiota and the Gut–Skin Axis

The intestine and the skin share a number of common features: they are border barriers between the external and internal environment, both possessing an epithelial surface of about 30 m^2^ that, in the epidermis, is mainly determined by the presence of hair follicles, apocrine/eccrine ducts, and sebaceous glands [4] (Figure 1).

The skin and the gut also share common neuroendocrine properties, driven by gut microbes that produce neurotransmitters such as acetylcholine and serotonin, able to stimulate, via the neural system, the secretion of hormones from specialized intestinal cells, which finally determine systemic and inflammatory effects also involving the skin [4] (Figure 2). Beyond these peculiarities, these districts also show a key role in the mediation of inflammatory conditions and immune development, since, starting from early childhood, they identify novel antigens daily to distinguish whether and how to tolerate them [5].

To this aim, and to guarantee the host’s allostasis and homeostasis, they cooperate in the production of anti-microbial agents and/or specific nutritional sources [6,7]. By selecting the exogenous and endogenous microbial colonizers that compete for the epithelial surfaces, they prevent the possible numerical prevarication and virulence of primary and/or opportunistic pathogens [1,4,5]. From the skin side, this task is also fulfilled by flaking, which, together with keratin and skin components such as the sebum, allows the epithelial renewal process and protection from weak acids/bases and various antigen types; conversely, from the intestinal mucosa part, the same is done by the mucin glycoproteic components within the epithelial villi [8,9].

Moreover, these two districts are not distinct; they are one the continuation of the other, intimately connected and perfused by the bloodstream. They are two faces of the same coin, “separated” on one side by the mouth and the perineum and, on the other, by the nasal pits and the skin pores, respectively, the “entry” and “exit” routes of the digestive and respiratory systems. 

Each one of them possesses a peculiar microbiota, mainly identified through the 16S ribosomal RNA (rRNA) gene sequencing, which finely orchestrates a two-way collaborative relationship [10,11]. A lot of evidence underlines this unequivocal connection and the involvement of the immune and endocrine systems. The skin microbial community is composed of a collection of microorganisms that, with the sole exception of the core microbiota that remains constant over time, vary during life, conditioned as they are by influencers such as hormones, anatomical distribution, pH, and hygiene habits, which determine their peculiar distribution in the diverse dry, moist, and sebaceous areas.

Among them, there are the Actinobacteria, which represent half of the residents, Firmicutes (mainly *Staphylococcus* and *Streptococcus* spp.), Proteobacteria, and Bacteroidetes bacterial phyla but also specific beta and gamma Human Papillomavirus (HPV) genera, *Malassezia* spp., archaea, and protozoa [4,12].

The gut, mostly the large intestine, is indeed partially colonized by the same Actinobacteria (with Bifidobacteria inducing anti-inflammatory Treg cell accumulation at the basis of the immune tolerance) and Firmicutes, both dominant (90% of gut microbiota). Proteobacteria, Bacteroidetes, Fusobacteria, Verrucomicrobia, DNA/RNA viruses, *Candida*, and protozoa are also key representatives [13]. The gut residents, whose distribution and abundance depend on the anatomical region, pH, and O_2_ values, also have a metabolic role, with the production of short-chain fatty acid (SCFA) bio-products: in fact, the acetate, butyrate, and propionate, which cooperate for the epithelial barrier integrity preservation, prevent or solve local and systemic inflammatory and immune effects [4,13,14,15,16] (Figure 3).

The number of microorganisms that inhabit the skin and the gut has been calculated to be 10^8 − 11^ and 10^14^, respectively, with at least 1000 and 500 species in each niche. Their balance is, throughout life, conditioned by the exposome, which includes endogenous and exogenous chemical, physical, biological, and social factors that influence their karyosome and that of one of the hosts [15,17].

In a eubiotic microbiota, there are mainly commensals or symbionts, but when a chronic selective pressure occurs, only the more resilient ones can survive, become dominant, and behave as opportunistic pathogens, thus increasing the host’s general susceptibility to endogenous and exogenous infections and immune derangement [2,4,11].

In this direction, the gut microbiota can highly impact skin microbial colonization, immunity, and health; most epidermal diseases come from gut dysbiosis: the research in support of the reciprocal cause–effect relationship between the two is one of the areas of greatest scientific debate [5].

One of the most accredited hypotheses to prove the link between them is how, starting from intestinal dysbiosis, a systemic skin T-cell activation and an anti-inflammatory cytokine and Treg cell function downregulation can happen, thus bringing about uncontrolled gut and skin inflammation, no longer adjustable by the immune response.

Moreover, as evidenced in chronic skin disease patients, when the intestinal barrier integrity is disrupted or highly compromised (leaky gut phenomenon), both at the physical (mucin) and functional (IAP alkaline phosphatase and antimicrobial proteins) levels, bacterial DNA, metabolites, bioproducts, and lipopolysaccharides (LPS) can directly access the bloodstream [18,19]. This favors their accumulation in distant sites, such as the skin itself, and the induction of the above-mentioned processes [18,19].

In line with this, evidence of the gut’s immuno-shaping properties on distant sites has allowed the commonly used term “inter-organ communication axis” to be coined, which explains the link between the gut and other organs such as the skin, brain, or lung [20].

In support of the centrality of the gut microbiota for skin health, is the in vivo analysis conducted in 2014 by Gueniche et al. [21] on a cohort of 32 subjects who, after being fed with an *L. paracasei* NCC 2461 strain, showed increased TGF-β blood levels, peaking at 62.0 pg/mL values at day 29 with respect to 47 pg/mL at day 1, which likely contributed to an improvement in the skin integrity, previously disrupted by a capsaicin-induced irritant stimulus, as assessed with the trans-epidermal water loss (TEWL) measure [22].

The skin resident microbiota is itself able to produce anti-inflammatory metabolites for local immune homeostasis and for protection from pathogens and physical and chemical damage [23,24].

The skin effects produced by UV radiation are a long-known story: they include the disruption of the exposed epidermal microbiota, photodamaging/aging, and proinflammatory and carcinogenic activities but also vitamin D synthesis and immunomodulation, innate immunity induction, and adaptive response suppression. These phenomena were observed by Patra and colleagues [25] in UV-irradiated healthy individuals, in which Toll-like receptors (TLRs), antimicrobial peptides (AMPs), and interleukin (IL)-1 family gene expression were significantly modulated.

More recent is the awareness of UV’s impact on gut microbiota: Ghali et al. [26], in 2020, demonstrated how UVB exposure to approximately 70% of a minimum erythemal dose of narrowband UVR (311 nm), thrice a week for a week by the end of winter, was able to increase the relative abundance of *Firmicutes* and *Proteobacteria* in the feces of 21 healthy subjects while reducing that of *Bacteroidetes*.

The same researchers showed a similar effect in a 6-week-old mice model: sub-erythemal UVR and 25-hydroxy vitamin D supplementation for 5 weeks modulated the abundance of the gut microbiota bacterial component by increasing its diversity and the host’s health [15].

On these bases, since UV-ray overexposure is mainly recognized for non-melanoma skin cancer (NMSC) trigger and progression, the step through which the gut–skin axis is conditioned by UV in the progression of this type of solid tumor should be short; nevertheless, until now, scarce literature and no research has been conducted in this direction to connect UV irradiation, gut dysbiosis, skin dysbiosis, and NMSC [27,28,29,30].

UV rays cause mutations in key genes such as those related to the transforming growth factor-beta (TGFβ) pathway, which normally oversees tumor cells thanks to its ability to inhibit keratinocyte overgrowth and affect tumor progression and whose expression is indeed context-dependent. These mutations prevent TGFβ-induced cell growth inhibition, favoring the typical genomic instability at the basis of the trigger and progression of cutaneous squamous cell carcinoma (cSCC) and melanomas.

Particularly, upon UV-B, TGFβ expression is enhanced in the keratinocytes of NMSC and in the cutaneous melanoma microenvironment, leading to vessel formation, inflammation, immune evasion, and metastasization [31].

Interestingly, the TGFβ signaling role in UV-induced photoaging and skin cancer is also connected to the modulatory key role of this highly conserved cytokine on the human host’s microbiota and the immune cell crosstalk, also through MMP2 and MMP9 release and collagen fibril damaging and elastosis [32].

## 3. Experimental Evidence of the Role of Microbiota in Skin Carcinogenic Processes

The role of the microbiota in skin carcinogenesis is an active area of research, even if it still remains unclear. The cutaneous microbiota has been found to play a role in maintaining skin barrier function, modulating the immune system, and defending against pathogens [33]. However, alterations in its composition and diversity have been associated with various skin diseases, including skin cancer, as demonstrated by many experimental studies.

*Staphylococcus epidermidis* (*S. epidermidis*) occurs naturally on healthy skin, where it plays a protective and antitumor role by activating the immune system to fight cancer cells and where it competes with other potentially harmful bacteria such as *Staphylococcus aureus* (*S. aureus*) [34].

In response to adverse external stimuli, the skin microbiota can become unbalanced, leading to a decrease in the presence of *S. epidermidis* and an increase in pathogenic *S. aureus*. Disturbances of the cutaneous microbiota are frequently observed in tumor patients undergoing radiotherapy, chemotherapy, and probiotics [28,29]. Several studies have demonstrated the association between *S. aureus* and increased susceptibility to skin cancer [35]. Specifically, the presence of *S. aureus* is strongly associated with SCC. Compared to healthy individuals, *S. aureus* is significantly more prevalent in the group of patients with SCC of the oral cavity [36]. The prevalence of *S. aureus* in the skin has also been found to be associated with cutaneous T-cell lymphoma [37]. Furthermore, *Pseudomonas aeruginosa* can also promote the growth of skin cancer cells [38].

Several mechanisms have been proposed through which the skin microbiota may influence skin cancer development: inflammation and immune modulation are the main involved. Dysbiosis, the imbalance in the skin microbiota, can lead to chronic inflammation, and this is a known risk factor for cancer development as it can promote DNA damage, cell proliferation, and immune dysfunction, each one of them contributing to carcinogenesis. The skin microbiota also interacts with the immune system, influencing its response. Particularly, dysbiosis can disrupt immune homeostasis and impair the immune response to cancer cells.

As is well known, the commensal and pathogenic skin microbiota regulates innate local immunity through keratinocytes, dendritic cells, mast cells, endothelial cells, fibroblasts, neutrophils, and macrophages [25]. As the first line of defense, the skin is constantly exposed to pathogen-associated molecular patterns (PAMPs) and damage-associated molecular patterns (DAMPs). After microbial stimulation, keratinocytes upregulate the production of antimicrobial lipids and antimicrobial peptides, and PAMPs and/or DAMPs are recognized by pattern recognition receptors (PRRs) [39]. Toll-like receptors (TLRs) are the major class of PRRs involved in detecting invading pathogens in the skin and play a vital role in the initial stage of cutaneous innate immune response by recognizing PAMPs and initiating immune signaling pathways. TLRs are expressed on many different cell types in the skin, including keratinocytes, melanocytes, and Langerhans cells in the epidermis. PRR activation triggers downstream immune signaling pathways, the activation of the innate immune system, and subsequent adaptive immune responses, leading to the clearance of invading pathogens [40].

Aberrant expression or the persistent activation of TLRs by pathogenic skin microbiota may promote chronic inflammation and can contribute to the generation and progression of many skin immune disorders, such as systemic lupus erythematosus, cryopyrin-associated periodic syndrome, and primary inflammatory skin diseases, including psoriasis, atopic dermatitis, and also cancer [41]. In particular, the relationship between various TLRs and skin cancer has been extensively studied [42]. Of the different TLRs, TLR4 is known to play a fundamental role in both skin inflammation and cancer. TLR4 is primarily known for its role in recognizing LPSs, a component of the Gram-negative outer membrane, but it can also recognize other endogenous and exogenous ligands [43]. The activation of TLR4 by Gram-negative bacteria and their LPSs and subsequent intracellular signaling pathways can turn on transcription factors such as NF-κB, IRF-3/7, and AP-1, which affect the expression of genes related to inflammation, cell apoptosis, survival, and differentiation [38]. The increased expression of TLRs has been observed in skin tumors. Several studies have investigated the role of TLR4 in SCC, particularly in the context of inflammation and the tumor microenvironment [39,43,44,45]. Chronic inflammation is believed to contribute to the development and progression of SCC. The overexpression of TLR4 was observed in SCC-affected skin compared to normal skin [44]. TLR4 activation increases the production of the proinflammatory cytokines TNF-α and TGF-β while inhibiting the anti-inflammatory IL-10. In SCC, TLR4 expression has been observed in both tumor cells and immune cells infiltrating the tumor. TLR4 signaling is associated with the activation of ERK, p38 MAPK, and MyD 88 pathways, which, in turn, activate downstream mediators, such as NF-kB, leading to the activation of genes encoding proinflammatory cytokines. The signaling pathways downstream of TLR4 were demonstrated to induce tumor cell proliferation, survival, invasion, and angiogenesis [45]. Many studies suggest that TLR4 signaling may have implications for tumor growth, metastasis, and the modulation of the immune response in SCC. Further research is needed to fully understand the complex interplay between TLR4 and SCC and to explore the potential of TLR4-targeted therapies in the management of SCC [46].

The overexpression of TLR4 has also been observed in malignant melanoma (MM), and TLR4 expression is negatively associated with recurrence-free survival [47,48]. Furthermore, studies have suggested that TLR4 signaling may be associated with the epithelial–mesenchymal transition (EMT), a process that allows epithelial cells to acquire invasive properties. EMT can, in fact, contribute to the metastatic process in melanoma [49,50,51]. Commensal skin microorganisms not only induce innate immune responses but also regulate the cutaneous adaptive immune system and the action of T lymphocytes. Recent studies have highlighted the importance of T-helper-type 17 lymphocytes and effector cytokines in cutaneous inflammation and microbiota-mediated skin carcinogenesis. In particular, IL-17 and IL-22 have been shown to be key factors in skin cancer progression, as they can induce cell proliferation in NMSC cells and promote the migration of human basal cell carcinoma (BCC) and SCC cell lines in vitro. Additionally, IL-17 and IL-22 stimulate tumor growth in mice injected with an SCC cell line, CAL27. However, further studies are needed to identify the specific skin microbiota that enhances the response of Th17 cells in the skin [52].

Furthermore, the microbiota can promote a tumor microenvironment through the action of microbial metabolites that can directly interact with tumor cells and/or influence carcinogenesis [53]. The interplay between microbiota-derived metabolites and the immune system is complex and multifaceted. While certain metabolites, such as short-chain fatty acids (SCFAs), have been associated with immunomodulatory effects that may have a protective role against cancer development, other metabolites, like certain secondary bile acids, have been linked to pro-inflammatory and potentially carcinogenic effects [54]. Deoxycholic acid and lithocholic acid can act as signaling molecules implicated in inflammation and carcinogenesis. Moreover, some studies have suggested that secondary bile acids, in particular deoxycholic acid, can induce DNA damage, promote oxidative stress, and alter cellular signaling pathways, potentially contributing to the development and progression of certain types of cancer in mouse models [55,56,57].

A reduced production by the cutaneous microbiota of the tryptophan metabolite, indole-3-aldehyde, is significantly associated with the degree of skin inflammation, which has potentially been described as promoting carcinogenesis [58]. The production of propionate and valerate by cutaneous *Propionibacterium acnes* (*P. acnes*) induces cytokine expression in response to TLR ligands, thereby promoting inflammation and carcinogenesis [59].

In the skin, some bacteria such as *S. aureus* and *P. acnes* can produce genotoxic compounds that directly interact with host DNA, leading to DNA damage and promoting carcinogenesis. Gram-negative bacteria produce cytolethal distending toxin, while the B2 phylogenetic group of *Escherichia coli* produces colibactin, and both genotoxins are capable of inducing double-strand breaks in host DNA. Moreover, *Streptococcus* and other bacteria generate reactive oxygen species, which can cause oxidative DNA damage, thereby increasing the risk of carcinogenesis [60].

Overall, while the precise role of gut microbiota in skin carcinogenesis is still being investigated, that of the epidermidis plays a complex and multifaceted role in local health and disease. Understanding the complicated interactions between the human skin microbiota and the body’s immune system may lead to new therapies for skin cancers and other diseases of this wide organ.

## 4. Microbiota and Melanoma

Bacteria and fungi can, through specific wall components, activate the host’s immune system, contributing to the maintenance of a pro-inflammatory and pro-oncogenic state; in melanoma, differences in microbiota composition based on the different stages may therefore underlie the dissimilar prognosis and disease course.

Studies conducted on melanoma animal models revealed some differences in microbe composition and microbial diversity with respect to normal skin [61,62]. In a porcine model, the 16s RNA sequencing demonstrated statistically significant differences in microbiota diversity and richness between melanoma tissue and healthy skin and between the fecal microbiota of MeLiM (Melanoma-Bearing Libechov Minipig) and control piglets [62]. In detail, the abundance of *Fusobacterium*, *Trueperella*, *Staphylococcus*, *Streptococcus*, and *Bacteroides* was the distinguishing feature of the melanoma microbiota, while *Bacteroides*, *Fusobacterium,* and *Escherichia-Shigella* were characteristics of the fecal microbiota of MeLiM [62,63]. Also, an abundance of *Prevotella copri*, *Clostridium IV*, *Holdemania*, *Anaerofustis*, and Saccharomycetales yeasts was demonstrated in patients affected by melanoma, with changes in the composition of gut microbiota between early-stage and invasive melanomas [64,65]. In detail, major differences (*n* = 180) in microbial communities were found comparing in situ and invasive melanoma; also, differences (*n* = 23) were observed between regressed and non-regressed melanomas [64]. Overall, the progression of melanoma from in situ to invasive forms is associated with a pauperization of the gut microbiota, with a decrease in alpha diversity. Moreover, bacterial species belonging to the order Clostridiales and species producers of butyrate were enriched in invasive melanoma. Even if the role of butyrate in cancer biology is still controversial [66], it has been demonstrated that this SCFA is involved in tumoral invasion through the activation of the EMT signaling pathway in human melanoma cells [67].

In the last decade, immunotherapy has revolutionized melanoma treatment, both in the adjuvant and advanced disease setting, greatly improving the prognosis of patients affected by this tumor. In this context, the microbiota controls the response and tolerability in patients treated with immunological checkpoint inhibitors (ICI). Indeed, antibiotic exposure has been related to worse prognosis in patients treated with ICI [68], even if a more recent study [69] on 169 melanoma patients treated with anti-PD1 did not show any association between antibiotic therapy, PFS, and OS. Also, a putative role of diet has been supposed. A recent study [70] demonstrated a correlation between dietary fiber intake and prolonged PFS in a large cohort of melanoma patients treated with ICI compared to a control group.

Recently, a clinical trial evaluated the safety and efficacy of the transplantation of responder-derived fecal microbiota together with anti-PD-1 in a small group of patients affected by PD-1–refractory melanoma [71,72]. This combination was well tolerated, with an objective response in 3/15 patients and a durable stable disease in three others. The authors reported a persistent microbiota perturbation in responder patients, which exhibited an increased abundance of taxa that were previously shown to be associated with response to anti-PD-1, together with increased CD8+ T cell activation and a decreased frequency of interleukin-8-expressing myeloid cells. Differences and the rate of change in microbial communities were evaluated using multidimensional Euclidean distance; even if the limited size of the sample did not allow the authors to reach a statistical significance, they report that Euclidean distance notably separated responder from non-responder. Also, the study published by Baruch et al. [73] demonstrated clinical responses (two partial and one complete) in anti-PD-1 refractory patients who received fecal microbiota transplantation (FMT) together with reinduction with anti-PD1. In this study, patients received an antibiotic pretreatment (neomycin plus vancomycin) to eradicate their native microbiota; it was noted that responder patients were characterized by a high relative abundance of *Ruminococcus* and *Bifidobacterium* species, previously described as favorable to immunotherapy, whereas *Clostridium* was the species most characterizing the microbiota of non-responsive subjects. Moreover, the post-treatment comparison between responders and non-responders revealed, in responders, a higher relative abundance of *Enterococcacee* spp., *Enterococcus*, and *Streptococcus australis* and a lower relative abundance of *Veillonella atypica*, with statistically significant differences. Recently, the results of a phase I multicenter clinical trial have been released [74] in which previously untreated patients with advanced melanoma received pembrolizumab or nivolumab together with healthy donor FMT. The objective response rate was 65%, with 20% complete responses. Longitudinal microbiome profiling demonstrated that all patients engrafted bacterial strains from their respective donors, with a similarity between donor and patient microbiomes that increased over time in responders. At present, several other phase I (NCT03772899; NCT03353402) and phase II (NCT04521075; NCT03341143) clinical trials that combine FMT with ICI administration are ongoing, with the aim of confirming these preliminary results. However, some issues about safety are still open, including the risk of bacteriemia and the selection of multi-drug-resistant pathogens.

Also, immune-related adverse events (irAEs) could be influenced by microbiota. In 2017, Chaput et al. [75] demonstrated, in 26 patients affected by MM treated with ipilimumab, that subjects with a baseline microbiota enriched by Firmicutes (*Faecalibacterium*) had a more favorable clinical response but also a more frequent occurrence of ICI-associated colitis. Moreover, a study conducted on 77 advanced melanoma patients treated with an anti-CTLA4/anti-PD1 combination demonstrated that the more severe irAEs were associated with a gut microbiome significantly (*p* = 0.009) enriched by *Bacteroides intestinalis* and *Intestinibacter barlettii* species [76]. Even ICI-associated colitis can be successfully treated with FMT, which is also capable of restoring a eubiotic intestinal microbiota, as described in Wang et al. [77].

## 5. Microbiota and Non-Melanoma Skin Cancer

NMSC represents the most common kind of tumor affecting human beings and is mostly represented among Caucasian patients. NMSCs include different clinical entities, often occurring in sun-exposed cutaneous sites in fair-skinned subjects. The most frequent NMSCs are represented by keratinocyte carcinomas, namely BCC and SCC, unless other types can also be seen, such as keratoacanthoma, Bowen’s disease, and their precursor actinic keratosis (AK). Both BCC and SCC arise from keratinocytes, but they present some pivotal differences, as they originate from the diverse cellular layers of the epidermis (keratinocytes from the basal layer in BCC, and the spinous layer in SCC), and, most importantly, they recognize different clinical features and behavior, with a moderate/strong aggressiveness and a tendency to metastasize in SCC, which is extremely rare in BCC [78]. AK is a very common cutaneous lesion affecting numerous patients worldwide, and it typically develops on sun-exposed sites and in the context of the so-called field cancerization (FC), i.e., a cutaneous area in which clinical and subclinical actinic damage coexist, possibly with DNA damage, with a high risk of developing AKs and SCCs. AK is commonly considered as an SCC precursor into a possible continuum towards an invasive carcinoma unless it is currently impossible to predict which AK will progress into the invasive form [79,80].

On the other side, it is well known that UV exposure, old age, pale skin, and immuno-suppression represent important factors in favoring the development of AK and, possibly, its transformation. In this regard, solid organ transplant recipients (SOTRs) have up to 250-fold higher risk of developing NMSC, mainly SCC, when compared to immunocompetent subjects, and this has been explained by several factors: immunosuppressant drug intake, which induces photosensitivity and favors cutaneous UV-mediated DNA damage; a decreased capability to counteract the carcinogenetic process; and, not least of all, microbiome changes [81,82].

When considering the microbiota’s role in NMSC’s development in AK, the role of the HPVs belonging to the beta genus (namely ßHPVs) must be carefully evaluated. ßHPVs are a part of the virota and are typically harmless in the general population unless in the presence of certain predisposing factors (i.e., immunosuppression), in which they can promote cutaneous carcinogenesis; such a process has been first described in Epidermodysplasia Verruciformis (EV) patients. EV is a rare, genetically inherited disease characterized by abnormal susceptibility to ßHPVs, which can induce the development of multiple NMSCs early in life through the suppression of the apoptosis of UV-damaged cells [83]. Whether EV has been formerly considered a natural model to investigate the effect of ßHPVs in skin carcinogenesis, more recently, the same process occurring in immunocompromised patients (either for HIV infection or upon chronic immunosuppressive drug intake) led to the coining of the term “acquired EV” [84,85]. On this basis, ßHPV genomes were repeatedly detected either in malignant skin lesions, in perilesional healthy skin, or in plucked eyebrows from SOTRs (up to 85% of the considered samples), and an association between the viruses and AKs/SCCs was also reported, thus validating their role in skin carcinogenesis [86,87]. Accordingly, the detection of ßHPV protein expression into different kinds of keratinocyte tumors collected from SOTRs demonstrated the active viral infection, despite it having been supposed that the virus might act with a “hit-and-run” mechanism, thus requiring its presence only in the initial stage of the carcinogenesis [88,89].

Another possible explanation for viral-mediated skin carcinogenesis was suggested by Strickley et al. [90] through a murine model in which mice harboring a specific immunity towards MmuPV1 were infected with that viral genotype, and, surprisingly, they were protected against UV-induced skin cancer with a CD8 T cell-dependent mechanism. Accordingly, they demonstrated the existence of an adaptative immunity towards HPVs in normal skin from healthy subjects, thus pointing out the possible use of T cell-based vaccines against commensal HPVs. On the other side, ßHPVs DNA has been detected not only in samples from malignant lesions but also in plucked eyebrows collected from healthy individuals; hence, the persistence of the same type of specific ßHPVs in time was found, and thus the exact role of such viruses in NMSC remains controversial [91,92].

Moreover, many studies have been focused on the role of *S. aureus* in cutaneous carcinogenesis, and, once again, the data are related to SCC and AK, while information on the other skin tumor types is scarce. Indeed, Kullander et al. [36] showed a strong association between *S. aureus* and some kinds of NMSCs (mainly SCCs and AKs) in their study considering both biopsy specimens and skin swabs. More in detail, they researched the DNA fragments in samples collected from patients complaining of NMSCs versus healthy subjects, and they were able to find a higher association in patients affected by SCCs (prevalence 29.3% in SCC specimens versus 15.7% in healthy skin), and in the enrolled population, such a percentage was higher for *S. aureus* than for HPVs. Similar data were subsequently confirmed in a longitudinal and cross-sectional study conducted both in immunocompetent and immunocompromised subjects presenting AKs and with a history of SCC, in which *S. aureus* was found to be associated with SCCs and AKs, thus validating its potential role in favoring AK’s progression to SCC [82]. A further study conducted on SOTRs analyzed both the skin and gut bacteriome and mycobiome with the aim of highlighting the differences between patients with a history of multiple SCCs and matched SOTRs without previous skin cancer. The authors detected a significant reduction in the microbiota diversity in the group with a history of SCCs, and the dysbiosis was speculated to be responsible for a decrease in SCFA-producing microorganisms and a subsequent proinflammatory status, thus favoring the development of skin neoplasms [93]. Furthermore, in immunocompetent subjects, dysbiosis was also associated with SCC and AK with a predominant detection of *S. aureus* and of *Ralstonia pickettii*. Interestingly, a peculiar shift from the detection of a relative abundance of *Cutibacterium acnes* and *S. hominis*, which were mainly detected in non-skin cancer samples, towards an increase in *S. aureus*’ presence in AK samples was found, and that change in microbiota was more evident in the case of AK’s progression into invasive SCC [94]. Subsequently, Madhusudhan et al. [95] confirmed that *S. aureus* is more abundant in AK and SCC, and this could be related to the capability of the bacterium to increase the expression of human ß-defensin-2 (hBD-2), an important factor able to promote the tumor growth. On the contrary, they observed a significant increase in *Streptococcus* detection in BCCs [95]. *S. aureus* can also induce a state of chronic inflammation through the production of a phenol-soluble-α modulin (PSMα), a virulence peptide able to favor the release of different pro-inflammatory cytokines (e.g., IL-1α, IL-36α, IL-6, IL8, TNF- α, IL-17), enzymes and cytolytic toxins (e.g., leucocidins, hemolysins, serine proteases), and lipoproteins (SACOL0486) and triggering a self-maintained inflammatory mechanism [96,97]. With this in mind, it can be stated that NMSC development might be promoted by multiple trigger factors represented by keratinocyte DNA damage due to UV exposure, microbiota changes (together with the derived oxidative stress process), and the subsequent chronic inflammatory state; these factors can act jointly and lead to a self-maintained process [30].

Conversely, it has been reported that some commensal skin bacteria might protect against NMSC development, and this is of great importance in terms of prevention and potential treatments in high-risk subjects. Indeed, the detection of a particular strain of commensal *S. epidermidis* was found to be associated with a protective effect against skin cancer development. This was explained by the capability of the bacteria to produce 6-N-hydroxyaminopurine (6-HAP), a chemical compound exerting antiproliferative activity against neoplastic cell lines [98]. Furthermore, while Wood et al. [82] reported in their study a significant association between *S. aureus* and AK/SCC, they failed to detect acne *bacterium* and *Malassezia* in skin cancer samples, and these two microorganisms were relatively more abundant only in perilesional but healthy sun-damaged skin. Accordingly, *Malassezia* has been previously reported to be decreased in SCC samples and its protective effect against *S. aureus*-mediated cutaneous carcinogenesis has been speculated [36,95]. On the other hand, it has been reported that photodynamic therapy (PDT) induced a decrease in Malassezia abundancy in the perilesional skin of NMSC patients [99,100] Moreover, a further protective effect mediated by *Corynebacterium striatum* has been shown in the past and it is based on the capability of the bacterium to induce a change from a pathogenic into a commensal behavior of *S. aureus* in the skin microbiome by decreasing its virulence factors [101].

Taken together, these data could lead to the development of possible therapies against NMSCs by acting on the microbiota, even simply at a topical level [102]. In this regard, some studies have reported the potential photoprotective effect of pre- and probiotics, either by oral or topical administration, in UV-irradiated skin [103,104,105]. When considering AK, the effect of topical FC-directed therapy (i.e., diclofenac gel) on microbiota has been recently investigated, thus showing a decrease in *S. aureus* together with an increase in *Corynebacterium* bacterial load after treatment completion in responder patients, while the opposite was found in non-responders, with a potential repercussion in terms of follow-up and the identification of lesions at high-risk to progress to SCC. More in detail, *Staphylococcus* was relatively more abundant before the treatment with a value of 35.2%, 19.8% at week 24 (end of therapy), and 22.8% after the completion of therapy (week 36), while the *Corynebacterium* was detected in 14.9% of responder patients at the end of treatment (week 24) and in 16.8% after 36 weeks, while the percentages in non-responders were 3.98% and 4.5%, respectively [106].

Additionally, Voigt et al. [94] reported a positive effect of immunotherapy administered in SKH1 hairless mice affected by UV-induced SCC and simultaneously treated with the oral or topical administration of microbes (i.e., *Akkermansia muciniphila* per oral intake, *P. acnes* topically applied). In their experiment, mice were also pretreated with broadband antibiotics to deplete their cutaneous microbiota; thus, the authors were able to demonstrate the potential activity of microbiota manipulation in boosting the immunotherapy effect against the growth and tumor burden in SCC therapy.

## 6. Microbiota and Other Cutaneous Tumors

### 6.1. Cutaneous T-Cell Lymphomas

The etiopathogenesis of cutaneous lymphomas, which are rare and heterogeneous neoplasms of the lymphoid compartment in which the skin represents the first site of involvement, remains incompletely clarified. It has been hypothesized that various triggering factors can facilitate chronic inflammation and subsequent neoplastic transformation, and among them, a role could be played by the microbiota.

In mouse models of cutaneous T-cell lymphoma (CTCL), the putative influence of microbiota has been supposed through an observation regarding a less severe disease in mice raised in germ-free conditions [107]. So, several studies focused on the possible distinctive characteristics of cutaneous microbiota in CTCL patients, and on dissimilarity between lesional and non-lesional skin. A study conducted on 20 patients affected by stage Ia-IIb mycosis fungoides (MF) [108] revealed differences in the abundance of ten bacterial species between MF patches and/or plaques and patient’s unaffected skin; in detail, *Streptomyces* sp. SM17, *Bordetella pertussis*, *Streptomyces* sp. PVA 94-07, *Methylobacterium oryzae, Serratia* sp. LS-1, *Burkholderia mallei*, *Enterobacteriaceae bacterium*, *Achromobacter ruhlandii*, *Pseudomonas* sp. A214, and *Pseudomonas* sp. st29 were significantly more abundant in healthy skin areas. Also, a pilot study [109] demonstrated a higher percentage of commensal *Staphylococcus spp* in CTCL patients compared with healthy volunteers and a higher relative abundance of *Corynebacterium* and decreasing trends in *Cutibacterium,* with differences correlated to the disease stage. These data were also confirmed by another larger study enrolling 39 patients [110]: significantly higher *Corynebacterium* spp. abundance was observed in lesional skin, with a median relative abundance of 11.8%, together with *Neisseriaceae* spp., whereas non-lesional skin was characterized by an increased abundance of *Sandaracinobacter* spp. and *Enhydrobacter* spp. In their study, these authors also demonstrated a correlation between different bacteria species and disease phenotype (erythroderma or patches or plaques) and symptoms (pain or pruritus), also corroborating the well-established observation relating to the *S. aureus* colonization in erythrodermic CTCL patients [111,112]. On the other hand, the abundance of *Staphylococcus* spp. decreases in CTCL patients undergoing narrowband UVB and responding to this treatment [113].

More interesting are the data obtained by Dehner et al. [114], who revealed the presence of *Bacillus safensis* in seven patients affected by MF. This bacterium, not evident on the skin of healthy control subjects, has been shown to be able to stimulate T-cell proliferation in vitro through the production of cytokines and growth factors and could confirm that microbes can act as trigger factors for tumorigenesis.

In 2021, a case-control study published by Hooper et al. [115] demonstrated a gut dysbiosis correlated with the disease stage in 38 CTCL patients, compared to age- and geographically matched healthy controls. In detail, CTCL patients were characterized by significantly lower α-diversity and by a loss of butyrate producers (i.e., *Bifidobacterium* and *Anaerotruncus,* and *Lactobacillus* in subjects with advanced disease). These observations also create the foundation for the possible use of FMT in the treatment of this neoplasia.

Also, in CTCL, a putative tumorigenic role could be played by viruses. Cutavirus, which is a recently discovered member of the Parvoviridae family, has been identified both in skin samples and in feces in patients affected by different cutaneous diseases, also including CTCL, while it is not detectable in healthy subjects [116,117,118]. However, the role of this virus remains to be clarified, also based on the work conducted by Bergallo et al. [119], who have not identified its presence in any of the 55 CTCLs tested and by Harkins et al. [109], which led a shotgun metagenomic sequencing analysis in six CTCL patients, demonstrating a low viral abundance without significant differences in comparison to healthy volunteers. 

### 6.2. Merkel Cell Carcinoma

Merkel Cell Carcinoma (MCC) is a rare skin tumor arising on sun-exposed sites, mainly in immunocompromised and elderly patients, and with aggressive behavior. Despite the possible origin of MCC from its related Merkel cell (located in the basal layer of the epidermidis, thus providing a mechanoreceptor function) still being under debate, the so-called Merkel cell polyomavirus (MCPyV) has been found in a high percentage of the corresponding tumor (ranging from 58% to 88% of cases, according to the different detection techniques), thus confirming its causal role in MCC development [120]. Indeed, it has recently been proposed that MCC can recognize different cellular origins: from dermal fibroblasts and from epidermal keratinocytes, and the difference between those target cells is due to MCPyV’s presence or not. More in detail, two MCC entities can be described: the virus-positive MCC-targeting fibroblasts, and the virus-negative MCC arising from the keratinocytes. MCPyV is a part of the human skin microbiota able to encode for two oncoproteins (LT: large tumor antigen; sT: small tumor antigen) that are implicated in carcinogenesis through viral integration into the host genome in dermal fibroblasts, the inhibition of pRb1, and the evasion of the immune response. Conversely, in virus-negative MCCs, the target cell is represented by the epidermal keratinocytes in which the DNA damage and somatic mutations are multiple, thus resembling those occurring in the case of chronic UV exposure [121].

Hashida and colleagues [122] have previously investigated the presence and viral load of MCPyV either in non-lesional sun-exposed or unexposed skin from six Japanese patients affected by MCC, thus performing a comparative sequence analysis of the virus collected from the different sites. Interestingly, they found that the viral genome isolated from the MCC harbored various mutations in the oncoproteins LT and/or sT, while in normal skin, a wild-type sequence was more commonly detected, pointing out the possible initiating UV effect in skin holding MCPyV.

Furthermore, MCPyV has also been detected in other neoplasms affecting the oral cavity and the gastrointestinal system, in lung, renal, prostate, and cervical cancer, as well as in skin samples collected from healthy individuals [123]. Regarding soft tissue neoplasms, some studies have reported the presence of MCPyV in different NMSCs, the majority represented by SCCs affecting immunosuppressed patients, but also in AK, atypical fibroxantoma, keratoacanthoma, Kaposi’s sarcoma, porocarcinoma, and dermatofibrosarcoma, despite the relatively low viral load and the lack of viral integration into the host genome having raised the question as to its possible coincidental rather than causal role in the genesis of such cancers [124,125,126]. Accordingly, serological evaluation was found to be positive in a high number of healthy subjects starting from infancy (45%) towards adulthood (60% to 81% depending on the considered age category), despite a higher antibody seroprevalence in MCC patients versus healthy controls (90.0% and 67.6%, respectively), thus suggesting that MCPyV belongs to the skin microbiota, and its role in promoting MCC needs to be further studied [127].

### 6.3. Kaposi’s Sarcoma

The pathogenesis of Kaposi’s sarcoma (KS) is notoriously related to the Kaposi’s sarcoma-associated herpesvirus (KSHV), also known as herpesvirus-8, which is one of the seven viruses classified as human carcinogens by the International Agency for Research on Cancer [128]. KSHV is a DNA virus that is capable of infecting not only endothelial cells, from which KS originates, but also monocytes and B cells, acting on cellular metabolism, upregulating the survival pathways, and stimulating angiogenesis and inflammation.

Together with the skin, the oral cavity represents a common site of involvement in KS patients, representing the first affected site in 20–60% of HIV-associated cases. Several studies demonstrated a change in the microbial diversity in HIV-positive patients [129], with a high prevalence of severe oral inflammation and periodontal disease, which in turn are associated with KS progression [130,131]. In detail, a reduction in the microbiota diversity has been documented, with an increase in *Firmicutes* and *Streptococcus* and a decrease in *Lactobacillales* and *Pasteurellaceae* [131]. It has been supposed that some bacterial products, such as butyrate, can promote viral replication and dissemination, resulting in KSHV reactivation [132,133]. Also, a study recently conducted on 29 KS patients [134] confirmed a diminution of the oral microbial diversity and the enrichment of specific bacteria in individuals coinfected by HIV and KSHV.

## 7. Potential Therapeutic Implications

The therapeutic effect of the human microbiota in skin cancer is an emerging area of research, and although it is still being explored, several potential avenues for therapeutic intervention have already been identified. The human microbiota has the potential to influence the development and progression of skin cancer and also has a therapeutic impact on it in several ways: (1) microbiota modulation, (2) immune modulation, (3) the production of anticancer metabolites, and combining microbiota-based therapies with conventional cancer treatments [59].

### 7.1. Microbiota Modulation

We have described how the microbiota acts as a crucial regulator of the tumor microenvironment, modulating tumor development and progression. Manipulating the composition and diversity of the skin microbiota may offer therapeutic benefits. Probiotics and prebiotics could be optimal therapeutic options due to their ability to regulate cutaneous and intestinal dysbiosis. It should be noted that restoration of the gut microflora may also have a beneficial effect on the skin, as disturbances of the gut microbiota may be associated with the development of skin tumors.

Prebiotics and/or probiotics can be used to promote the growth of beneficial bacteria while suppressing the overgrowth of potentially harmful ones. This approach aims to restore microbial balance and enhance the skin’s natural defense mechanisms against cancerous cells [135].

Probiotics such as *Lactobacillus* and *Bifidobacterium* genera are live microorganisms that can exert beneficial effects not only on the well-studied and documented gut microbiota but also on the skin. They can inhibit the growth of pathogenic microorganisms and promote an anti-inflammatory phenotype of the epithelium [135]. The oral and topical administration of probiotics appear to be effective for the treatment of various inflammatory skin diseases and dermatological conditions, including atopic dermatitis, acne, and psoriasis, and are also showing a promising role in wound healing and skin cancer [136,137,138,139].

UV radiation is a strong environmental risk factor for skin cancer. Oral probiotics have been found to have a photoprotective effect by inhibiting skin tumor growth. Strains of *S. epidermidis* produce 6-HAP, a molecule that inhibits DNA synthesis and exhibits antitumor activity, as well as the ability to suppress de novo growth induced by UV irradiation, suggesting a mechanism by which commensal skin bacteria may help protect the host from skin neoplasia [98]. The cell wall of lactobacilli possesses lipoteichoic acids (LTAs), which are molecules with immunomodulatory properties. The oral administration of LTAs can delay the development of UV-induced tumors [101], while the topical application of *Lactobacillus plantarum* reduces colonization of the skin by *P. aeruginosa*, a Gram-negative opportunistic pathogen involved in carcinogenesis, as described above [139].

Although no studies have tested the clinical effect of topical probiotics on skin cancer thus far, topical probiotics, rather than oral, should be more effective in directly targeting the skin microbiota in patients with skin cancer, thanks to the direct modulation of the cutaneous and intratumoral microenvironment, as described earlier in this review.

### 7.2. Immunomodulation by Microbiota

The skin microbiota interacts with the immune system, and its modulation can influence the immune response. Certain microbial species or metabolites produced by the microbiota can enhance immune surveillance against cancer cells or regulate inflammation, which plays a critical role in tumor development. Harnessing these immunomodulatory properties could help improve therapeutic outcomes in skin cancer.

Supplementation with highly active strains of *Lactococcus* and *Lactobacillus* has demonstrated immunomodulatory, anti-inflammatory, antiallergic, and antitumor properties. *L. lactis* has been shown to produce anti-inflammatory compounds, such as bacteriocins and bioactive peptides that can help to reduce inflammation in the body. These compounds can inhibit the production of pro-inflammatory cytokines, modulating both Th1 and Th2 immune cell responses [140]. This ability to regulate T-cell activity is essential for maintaining immune balance and preventing excessive immune responses, inflammation, and carcinogenesis. Specifically, the administration of *L. lactis* in the form of fermented milk improved the immune system by upregulating innate and acquired immune responses through the production of cytokines such as IL-4, IL-12, IL-6, IFN-g, and TNF-a by Th1 and Th2 lymphocytes [140]. Metabolites of lactic acid bacteria, such SCFAs, exopolysaccharides, and bacteriocins have promising anticancer potential. They have been shown to promote the development and function of Tregs, which help maintain immune tolerance and prevent autoimmune responses, interacting with immune cells such as macrophages and dendritic cells and leading to their activation. This activation may improve the immune system’s ability to detect and respond to pathogens and cancer cells, promoting an effective immune response [141]. LTA isolated from *L. plantarum* has been reported to have beneficial effects on skin health and to prevent skin diseases. In a recent study, LTA was reported to inhibit melanogenesis in B16F10 melanoma cells and exert anti-photoaging effects on human skin cells by regulating the expression of matrix metalloproteinase [142].

### 7.3. Bacterial Anticancer Metabolites

It is known that some bacteria, even within the skin microbiota—for instance, certain strains of *Staphylococcus*—have been found to produce metabolites with potential anticancer properties, suggesting a potential therapeutic benefit in cancer treatment. Streptozotocin, a natural compound produced by the bacterium *Streptomyces achromogenes*, is commonly used as a chemotherapeutic agent in the treatment of pancreatic neuroendocrine tumors. Rapamycin is another natural metabolite produced by the bacterium *Streptomyces hygroscopicus*. It has immunosuppressive and anticancer properties and is used in cancer treatment, especially in combination with other drugs. Also, bleomycin is an antibiotic produced by *Streptomyces verticillus*, widely used in cancer chemotherapy, especially in the treatment of SCC [143].

The development and investigation of bacterial metabolites as potential cancer therapeutics are ongoing, and researchers should continue to explore their mechanisms of action and potential applications in skin cancer treatment.

### 7.4. TLR-Targeted Therapy

We described that when Gram-negative bacteria infect the host, the LPS component of their cell wall highly activates TLR4. The activation of TLR4 by LPS leads to a signaling cascade that triggers the production of pro-inflammatory cytokines, chemokines, and other immune mediators. This robust immune response is critical for the host’s defense against invading pathogens. However, the dysregulated or excessive activation of TLR4 by LPS can lead to harmful effects, such as excessive inflammation and tissue damage. The prolonged activation of TLR4 has been associated with various inflammatory diseases, including sepsis, inflammatory bowel disease, autoimmune conditions, and cancer [144].

Targeting TLR4 signaling may hold promise for managing inflammatory conditions and controlling bacterial infections caused by Gram-negative bacteria. Research in this area continues to provide valuable insights into the complexities of innate immunity and its role not only in host defense but also in inflammatory diseases and cancer.

Recently, it was found that the topical application of a TLR4 inhibitor, resatorvid, can reduce the size and number of tumors in a mouse model of UV-induced skin tumorigenesis. The topical application of TLR4 inhibitors could therefore, in the future, be used to block TLRs in UV-induced NMSC [145].

In a human study, G100, a TLR4 agonist, has been shown to exert antitumor responses and enable tumor regression in patients with Merkel cell carcinoma, showing acceptable safety and potential clinical application [146]. Finally, imiquimod cream treatment, a TLR7 agonist, is considered an effective therapeutic option for SCC, BCC, CTCL, and lentigo MM. The suggested anti-tumor effect of imiquimod is the activation of Th17/Th1 cells and cytotoxic T lymphocytes by TLR7 [147].

### 7.5. Antibiotics

We described above that *Staphylococcus* species are a group of bacteria that naturally occur on the skin surface and are considered part of the normal skin microbiota. *Staphylococcus* is one of the predominant genera in the skin microbial community. While *S. epidermidis* is considered a beneficial member of the skin microbiota, *S. aureus* can be more opportunistic and cause skin infections and skin cancer. The *S. aureus* eradication in CTCL patients has been demonstrated to be associated with clinical improvement [148,149,150]. Moreover, in patients that ameliorate after systemic poly-antibiotic therapy, a decrease in cell proliferation and in the expression of interleukin-2 receptor (IL2R)-a and tyrosine-phosphorylated STAT3 (pYSTAT3) was observed [149,150]. However, the effective long-term utility of antibiotic therapy needs to be clarified, since *Staphylococcus* can recolonize and there is a risk of selecting resistant strains.

In summary, combining microbiota-based therapies with conventional cancer treatments such as surgery, radiation, chemotherapy, or immunotherapy could potentially improve treatment efficacy. Combining therapies that target both cancer cells and the microbiota could create synergistic effects that lead to better treatment outcomes; moreover, understanding the complex interactions between the human skin microbiome and the body’s immune system could lead to new therapies for skin cancer and other skin diseases.

## 8. Conclusions

The skin microbiota plays an important role in maintaining skin health and protecting the body from dangerous pathogens. It helps to keep the local pH balance, prevents the growth of harmful bacteria, and promotes the production of natural antimicrobial peptides. The composition of the skin microbiota can vary depending on factors such as age, gender, ethnicity, and lifestyle. In addition, changes in the skin microbiota can be influenced by environmental factors such as diet, hygiene practices, and exposure to UV and pollutants.

The emerging knowledge about the involvement of microbiota not only in inflammatory processes but also in cell differentiation, proliferation, and migration phenomena also raised awareness of its potential role in cancer. This could be of even greater importance in the skin because of the possible direct interaction between microbiota and environmental carcinogens, such as UV radiation. The characterization of the skin microbiota could be helpful for identifying patients at greater risk of developing skin neoplasms or also have a prognostic value, defining subcategories of patients more responsive to specific treatments or more prone to treatment-related adverse events. More interestingly, restructuring the patient’s microbiota, even through the FMT, could improve the activity of the immune system and enhance the response to antineoplastic treatments.

However, several issues remain unresolved, including differences in sampling and processing methods, which can impair the results obtained in various studies; the influences of concomitant pharmacological treatments and lifestyle habits; and the safety of FMT procedures.

Moreover, the putative role of other microorganisms, including fungi and protozoa, still remains to be fully clarified. Further large case studies will therefore be essential, as will translational clinical trials.

## Figures and Tables

**Figure 1 microorganisms-11-02386-f001:**
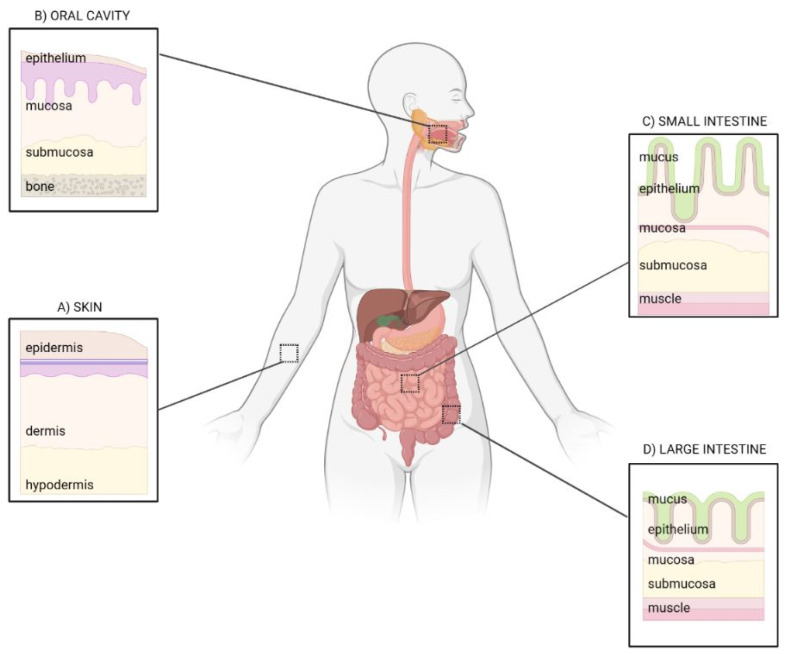
Epithelial barriers: from skin to gut. Common and specific traits of the skin (**A**), oral cavity (**B**), small (**C**) and large intestine (**D**). Image created with BioRender.com.

**Figure 2 microorganisms-11-02386-f002:**
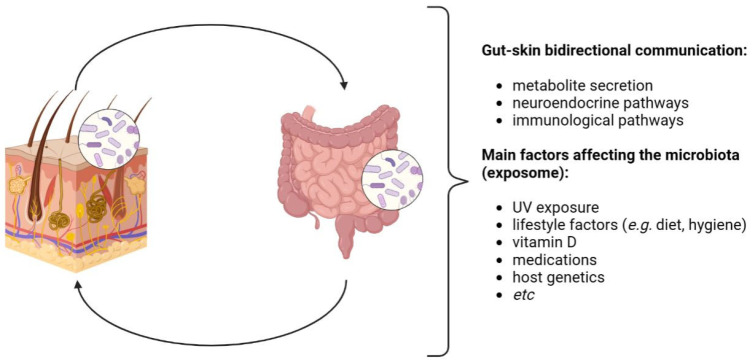
Gut–skin axis (GSA). The GSA is bidirectional: it is involved in the host’s homeostasis through immunological and neuroendocrine pathways. Gut and skin microbiota eubiosis is affected by several factors, generally recognized as the exposome. Image created with BioRender.com.

**Figure 3 microorganisms-11-02386-f003:**
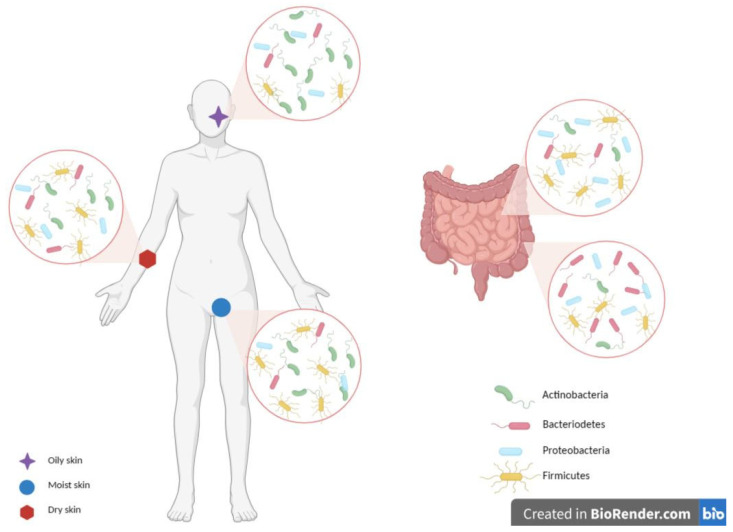
Skin and gut microbiota composition. The skin and gut are mainly colonized by *Firmicutes*, *Actinobacteria*, *Proteobacteria*, and *Bacteroidetes*, with a peculiar composition for each single niche. Moist sites are colonized mainly by *Firmicutes* and *Actinobacteria*, oily ones by *Actinobacteria*, while dry areas by diverse microbial populations. Gut microbiota differs between the small and large intestines. *Firmicutes* and *Proteobacteria* are the most dominant phyla in the small intestine, while *Bacteroidetes* dominate the anaerobic environment of the large intestine. Image created with BioRender.com.

## Data Availability

Data sharing not applicable.

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
