# Peer review of "Role of the Microbiota in Skin Neoplasms: New Therapeutic Horizons"

_microorganisms, 2023, doi:10.3390/microorganisms11102386_

Round 1

Reviewer 1 Report

The review article is very well structured, and no such changes are required, I am agreed to be published in the current version, however, I think this article needs to be cited due to its relevancy to the topic "https://doi.org/10.3390/ph16010087"

Author Response

Rev 1:

The review article is very well structured, and no such changes are required, I am agreed to be published in the current version, however, I think this article needs to be cited due to its relevancy to the topic "https://doi.org/10.3390/ph16010087"

We would like to thank this Reviewer for the enormous interest shown in our work. The bibliographic reference suggested has been added (Ref 34), and the numbering changed accordingly.

Reviewer 2 Report

The journal allows abstracts of up to 200 words, so I believe that authors can improve the abstract, highlighting the key points of the research.

Lines 41-45: Standardize formatting.

Line 89-90: Candida (and other genera and species) should be italicized. Check the entire manuscript.

Although the authors have brought an interesting qualitative analysis, I believe that the review lacks a quantitative analysis. What is the percentage of changes in microbiome in each case, with or without pathologies? The analysis of each article used should be more in-depth.

Reference 1 is not in the reference list.

Moderate editing of English language required.

Author Response

Rev 2: 
We would like to thank the Reviewer for the interest shown in our work and for suggestions. 
The journal allows abstracts of up to 200 words, so I believe that authors can improve the abstract, highlighting the key points of the research.
The abstract has been modified accordingly. The word count is now 202.
Lines 41-45: Standardize formatting.
Done
Line 89-90: Candida (and other genera and species) should be italicized. Check the entire manuscript.
Done
Although the authors have brought an interesting qualitative analysis, I believe that the review lacks a quantitative analysis. What is the percentage of changes in microbiome in each case, with or without pathologies? The analysis of each article used should be more in-depth.
The more significant articles were analyzed more deeply, as suggested.
Reference 1 is not in the reference list.
The reference 1 (Mahmud, M.R.; Akter, S.; Tamanna, S.K.; Mazumder, L.; Esti, I.Z.; Banerjee, S.; Akter, S.; Hasan, M.R.; Acharjee, M.; Hossain, M.S.; Pirttilä, A.M. Impact of gut microbiome on skin health: gut-skin axis observed through the lenses of therapeutics and skin diseases. Gut Microbes 2022, 14(1):2096995.), cited at line 31, is in the reference list.

Comments on the Quality of English Language
Moderate editing of English language required
We edited the manuscript and inserted some corrections in the English language, as required.

Reviewer 3 Report

Skin cancer is the most common type of cancer with an increasing prevalence worldwide. While ultraviolet radiation is a well-known risk factor, there is emerging evidence that the microbiota may also contribute. In recent years, the human microbiota has become a topic of great interest, and its association with inflammatory skin diseases (i.e., atopic dermatitis, acne, rosacea) has been explored. Little is known of the role of microbiota in skin cancer, but with the recognized link between microbial dysbiosis and inflammation, and knowledge that microbiota modulates the effect of UV-induced immunosuppression, theories connecting the two have surfaced. In this paper, the authors provide a comprehensive review of the key literature on human skin microbiota and skin cancer (i.e., non-melanoma skin cancer, melanoma, cutaneous T cell lymphoma). Also, mechanistic perspectives as to how our microbiota influence skin cancer development and treatment are offered.

The past decade has witnessed groundbreaking advances in the field of microbiome research. An area where immense implications of the microbiome have been demonstrated is tumor biology. The microbiome affects tumor initiation and progression through direct effects on the tumor cells and indirectly through manipulation of the immune system. It can also determine response to cancer therapies and predict disease progression and survival. Modulation of the microbiome can be harnessed to potentiate the efficacy of immunotherapies and decrease their toxicity. In this review, we comprehensively dissect recent evidence regarding the interaction of the microbiome and anti-tumor immune machinery and outline the critical questions which need to be addressed as we further explore this dynamic colloquy.

I congratulate the authors for the information provided in this manuscript. The topic addressed is very important and fits the requirements of the journal. To be considered for publication, there are still some details to be corrected and clarified. In the following I will detail my requests for the authors of the manuscript:

Lines 26-29: I recommend inserting references.

Lines 31-33: I recommend inserting references.

Lines 69-72: I recommend inserting references.

Lines 92-96: I recommend inserting references.

Line 99: to be modified, from Malassezia fungal spp...in Malassezia spp.

Lines 101-105: the names of the genus of microorganisms are written in italics.

Lines 114-121: the names of the genus of microorganisms are written in italics.

Lines 123-124: I recommend inserting references.

Lines 136-139: I recommend inserting references.

Line 146: to write what LPS means and the abbreviation in parentheses. It was written for the first time in manuscript.

Lines 165-169: I recommend inserting references and the names of the genus of microorganisms is written in italics.

Line 200: to modify in the manuscript, S. epidermidis...like this: Staphylococcus epidermidis (S. epidermidis). Likewise in the case of S. aureus (line 202).

Line 208: what the abbreviation SCC represents, is not explained in the manuscript.

Lines 217-221: I recommend inserting references.

Lines 243-244: I recommend inserting more references.

Line 291: what the abbreviation SCFAs represents, is not explained in the manuscript.

Line 303: to modify in the manuscript, P. acnes ... like this: Propionibacterium acnes (P. acnes).

Lines 330-333: I recommend inserting more references.

Line 463: what the abbreviation SOTRs represents, is not explained in the manuscript.

Line 503: what the abbreviation AK represents, is not explained in the manuscript.

Line 659: the name of the genus of microorganisms is written in italic.

Few references have been made to the role of fungi, protozoa, and other microorganisms in skin tumors in humans. Please insert in the manuscript other examples with microorganisms that can influence the evolution or induce a tumor process.

Author Response

We thank the reviewer for his interest and favorable comments on our work, and for his thorough revision, which allowed us to improve the paper.
The requested changes have been made, and highlighted in yellow in the new version.

In detail:
References have been inserted where required (please pay attention to the fact that the changes in the new version of the work result in a mismatch of the lines compared to the previous version). Also, two new references have been added. The numeration has been changed accordingly. 

Where necessary, the characters have been reported in italics.

Where necessary, abbreviations have been explained extensively in the text (e.g. lipopolysaccarides - LPS; short chain fatty acids SCFA). The extended definition of SCC (squamous cell carcinoma), AK (actinic keratosis) and SOTRs (solid organs trapiant recipients) was already present in the text.

Some references to Malassezia and Candida were already present in the text, but as requested by the reviewer, the paragraph dedicated to NMSC has been expanded with some references to the potential role of fungi in the pathogenesis of this pathology, with the addition of 2 new bibliographical references . The paragraph relating to the conclusion has also been modified accordingly.

We hope that in this form the manuscript responds to the requests of the reviewer, who we thank for his support in improving our work.

Round 2

Reviewer 2 Report

I consider the manuscript suitable for publication

Author Response

We thank the reviewer for this positive feed-back, and for his/her interest in or work

Reviewer 3 Report

The authors responded to all the comments and fixed the identified deficiencies.
